# The Downstaging Concept in Treatment-Resistant Depression: Spotlight on Ketamine

**DOI:** 10.3390/ijms232314605

**Published:** 2022-11-23

**Authors:** Alina Wilkowska, Wiesław Jerzy Cubała

**Affiliations:** Department of Psychiatry, Faculty of Medicine, Medical University of Gdańsk, ul. Dębinki 7, 80-211 Gdańsk, Poland

**Keywords:** treatment-resistant depression, difficult-to-treat depression, staging, ketamine, downstaging, neuroplasticity, immunomodulatory effect, stress resilience

## Abstract

Treatment-resistant depression is a pleomorphic phenomenon occurring in 30% of patients with depression. The chance to achieve remission decreases with every subsequent episode. It constitutes a significant part of the global disease burden, causes increased morbidity and mortality, and is associated with poor quality of life. It involves multiple difficult-to-treat episodes, with increasing resistance over time. The concept of staging captures the process of changes causing increasing treatment resistance and global worsening of functioning in all areas of life. Ketamine is a novel rapid-acting antidepressant with neuroplastic potential. Here, we argue that ketamine use as an add-on treatment of resistant major depressive disorder, based on its unique pharmacological properties, can reverse this process, give hope to patients, and prevent therapeutic nihilism.

## 1. Introduction

According to the European Brain Council, the one-year cost of mood disorders in Europe in 2010 was 113.4 billion euros [1]. The costs of treatment-resistant depression are even higher, due to more hospitalizations, more outpatient visits, and the cost of medication [2]. It has been estimated as 40% higher than the medical cost of major depressive disorder (MDD) [3]. The burden of treatment-resistant depression (TRD) includes suicidality, but also non-suicidal mortality, which constitutes the largest part of increased mortality in mood disorders. The three main causes are cardiovascular diseases, respiratory diseases, and cancer [4]. Data from the European cohort confirm the low quality of life, reduced work ability, and higher healthcare resource utilization due to these conditions [5,6].

A Sequenced Treatment Alternatives to Relieve Depression (STAR*D) trial revealed that according to the Quick Inventory of Depressive Symptomatology—Self Report (QIDS-SR16), the overall remission rate was 67%. For the first step, it was 36.8%, for the second, 30.6%, 13.7% for the third step, and 13% for the fourth step. Patients who required more treatment steps had a higher relapse rate during follow-up [7]. Based on the analysis of STAR*D results, it was suggested that due to such factors as publication bias, patient selection, and failure to report negative results, the estimated efficacy of antidepressants can be even lower [8]. According to the already mentioned Treatment-Resistant Depression Cohort in Europe non-interventional cohort study, 55% of patients experienced two treatment failures and 30% had three unsuccessful drug trials. After 6 months, 73% of patients did not show a response. After 1 year, 19.2% were remitted and 69% showed no response; moreover, 33% of patients who at months 6 achieved remission deteriorated and did not reach remission criteria anymore. Unfortunately, the authors also found that after 12 months only 40% of patients had their treatment modified and 60% were still on the same medication as at the study baseline [9]. Increasing relapse rates and a declining response to treatment with every episode confirm the progressive nature of MDD.

The European Group for the Study of Resistant Depression (GSRD) distinguished major risk factors for TRD: symptom severity, suicidal risk, a higher number of lifetime depressive episodes, and comorbid anxiety disorder [10]. 

Here we present a narrative review of recent literature on the clinical description and pathophysiology of TRD, and the evidence of the potentially transformative ketamine treatment effect. First, we present definitions of treatment-resistant depression (TRD) and difficult-to-treat depression (DTD) and describe staging models. In the second part, we concentrate on studies investigating neuroplasticity in TRD and the potential effect of ketamine on this process. Finally, we propose the concept of downstaging as an effect of ketamine treatment.

## 2. Results

### 2.1. Clinical Definitions

#### 2.1.1. Treatment-Resistant Depression

The official consensus on the definition of TRD (treatment-resistant depression) is still lacking, but based on systematic analysis, the most used definition is a lack of response after two to three adequate antidepressant treatment trials, irrespective of the drug class, plus an adequate trial of psychotherapy [11]. Depression that did not respond to three or more antidepressants and ECT (electroconvulsive therapy) is called treatment-refractory [12]. The concept of TRD came from the clinical observation of patients who do not respond to treatment, and it is unquestionably clinically relevant, although very difficult to define the phenomenon. The main obstacles in the way of a clear definition are the heterogeneity of the group, different levels of resistance, psychiatric and somatic comorbidities, age of onset, and the number of episodes. There is also no clear distinction between response, partial response, and non-response, and no consensus on what is an ‘adequate’ antidepressant trial, including outcome measures, AD class, and psychotherapy status [13]. Other factors such as spontaneous evolution, the placebo effect, and patients’ personal beliefs also need to be considered when treatment outcome is analyzed [14]. A recent systematic review and meta-analysis of studies on concomitant treatment used the term ‘early-stage’ TRD, which describes non-response to one adequate pharmacological or psychological therapy [15]. This perspective allowed the authors to include more studies for analysis. Concurrently, the authors state that it is not known if ‘early-stage’ TRD progresses into the next stages of the disease. Inconsistencies involved in TRD definition concern not only clinical practice but also research [11,16]. Aspects like pseudo-resistance, non-compliance, and poor tolerability further complicate defining resistant depression [17].

#### 2.1.2. Difficult-to-Treat Depression

A more practical, real-life approach is presented by the concept of Difficult-to-Treat Depression (DTD) proposed by Rush [18] and developed later in the form of an international consensus guideline [19]. The purpose of this approach was to define a practical basis for allowing decisions on when to re-evaluate patients’ treatment. It postulates that sometimes remission is not possible, but there is still a place for optimal symptom control when psychosocial stressors and coping strategies are defined [18,19]. DTD nomenclature has a smaller stigma potential than TRD, gives more hope, and describes depression as difficult but not impossible to treat. It emphasizes clinicians’, but also patients’ responsibility for the treatment outcome and involves the patient in the decision-making process. Unlike TRD, it considers patients’ perspectives and quality of life apart from clinicians’ evaluation. It also underlines the need to engage the family and caregivers in the healing process. Such an approach resembles the management of chronic diseases, such as heart failure, where the main goal is optimal symptom control, and where strategies such as psychotherapy, diet, self-help, and physical exercise are considered very useful [18,20]. Rush et al., in their recent paper, further develop the concept of DTD and recognize three issues to be addressed in clinical research involving this group of patients. One is defining and distinguishing subtypes of this heterogeneous group. The second problem is the assessment of outcomes. The authors point out that the traditional perspective used in trials with treatment-responsive patients is not optimal for the DTD group. This approach is based on short-term effects and does not focus on the durability of DTD, its side effect burden, and daily function. Another challenge is developing a clinical trial design that would allow generalizability and making a causal inference. The authors underline the need to further develop a taxonomy of DTD considering specific clinical features such as oversensitivity to medications. They also suggest a multidimensional approach to DTD clinical factors such as the course of the disease, family history, previous therapies, comorbid conditions, early life trauma, and treatment adherence. This way of describing patients would allow us to distinguish subgroups of patients which could later be investigated in clinical trials. The authors suggest that outcome assessment should also include evaluation of such aspects as symptom dynamics, functioning, general health, quality of life, costs, and other ones [21].

#### 2.1.3. Staging

Chronic diseases of a progressive nature are often described with the use of staging. The first attempt at describing resistant depression derived from oncology and laid the ground for developing staging TRD models [22]. The above-mentioned definition of early-stage TRD corresponds with Stage I TRD according to Thase and Rush’s model [15]. It has been suggested that, unlike in oncology, where the absence of the disease is defined by the lack of neoplastic cells, depression has no categorical indicator of disease activity. Nevertheless, inventing staging models is still a breakthrough in the understanding of TRD, helping to find the best strategy at a certain point of the disease continuum [18]. Staging models describe the level of resistance in TRD and reflect a dimensional approach. They consider the number and duration of treatments, different modalities, and the severity of symptoms. According to the recent systematic review, five staging models of TRD have been described. The authors point out the strengths and weaknesses of all models and analyze the definitions of TRD used in each of them. They conclude with crucial suggestions for future development such as measuring the strength of treatment, including the strength of augmentation strategies, which could give a clear indication of when to switch to an antidepressant or add an adjunctive agent. They also propose giving higher scores to treatments such as ketamine and ECT. Another suggestion is to develop guidelines for patients who do not respond to ECT or ketamine [23].

The models with the most evidence are the Maudsley Staging Model (MSM) and the more recent Dutch measure for the quantification of treatment resistance in depression (DM-TRD). MSM is the first model to include disease characteristics, such as the duration and severity of the current episode, although it does not allow for assessing the level of resistance in the current episode [23]. The Dutch model is more detailed and includes the response to psychotherapy, uses ratings for functional impairment and psychosocial stressors, and considers the presence of comorbidities [24,25]. An example of implementing the idea of staging in treatment guidelines can be found in the National Institute for Clinical Excellence (NICE) 2009 TRD guideline which is based on the stepped care approach [26]. Day et al. conducted a study to retrospectively assess the practical use of this guideline in 178 patients with TRD. They found treatment gaps between guidelines and the reality of care in this group of patients. The first one was the long delay in starting treatment, the second was poor access to psychotherapy, and the third was the delay in antidepressant medication change after 4–8 weeks. Another one is not starting adjunctive therapies such as lithium and atypical antipsychotics after two unsuccessful antidepressant trials, which again confirms the great need for improvement in the treatment of TRD. The authors also point out the lack of guideline standardization concerning the sequence, the number of steps, and the assessment of the best moment to go to another step [27]. The progressive course of TRD is presented graphically on Figure 1.

Inventing new, more effective strategies acquires an understanding of processes taking place in the brain in the course of TRD. It would not be possible without taking a closer look at neuroplasticity, specifically the neuroplastic processes which occur in the adult brain. 

### 2.2. Neuroplasticity

#### 2.2.1. Adult Neuroplasticity

Neuroplasticity describes processes that allow the brain to adjust to changing environmental factors. In an adult, brain neuroplasticity involves structural and functional synaptic changes such as the strengthening, formation, and elimination of synapses [28,29]. It starts with the neuronal activation of signalling pathways including extracellular signal-regulated kinase 1 and 2 (ERK 1/2) and the adenosine 3’,5’-cyclic monophosphate (cAMP)-response element-binding protein (CREB), which causes the release of neurotrophic factors such as brain-derived neurotrophic factor (BDNF) and vascular endothelial growth factor (VEGF). Neurotrophic factors affect transcription and as a result, cause structural changes such as the formation of dendritic spines and increasing synaptic strength [28]. Factors including stress and depression are associated with reduced synaptic connectivity and the atrophy of neurons. 

Brain-derived neurotrophic factor plays an important role in neuroplasticity and the pathophysiology of depression. BDNF expression is lower in depressed patients and antidepressant treatment can enhance it [30]. Studies on suicide victims and post-mortem studies on depressed patients show decreased levels of BDNF and tyrosine receptor kinase B (TrkB) in the cerebral cortex and suggest that it contributes to a reduced volume of the prefrontal cortex (PFC) and hippocampus, as well as to the loss of synapses in the course of depression [31].

#### 2.2.2. Neuroplastic Changes in Treatment-Resistant Depression

There is evidence for increased BDNF levels in treatment-resistant patients, which are hypothetically due to antidepressant use [32]. Evidence shows that BDNF is necessary to achieve the antidepressant effect [31]. BDNF expression and its receptor, tyrosine receptor kinase B (TrkB), activity are needed for the antidepressant action of traditionally acting (TAAD) and rapid-acting antidepressants (RAAD) [30]. It is also critical for neuroplasticity, which allows the brain to adjust to stressors and rewards [28]. BDNF expression is influenced by treatment. 

Traditionally acting antidepressants increase BDNF expression as an effect of long-term administration. Electroconvulsive therapy causes rapid BDNF release in PFC and the hippocampus [31]. Studies on ketamine show that it rapidly increases BDNF in the hippocampus even after a single dose [33].

Neuroplasticity is investigated in imaging studies on a functional level where connectivity changes are assessed, or on a structural level where grey matter volume is measured. Studies show that such changes are related to symptoms of depression and that antidepressants can reverse them [31]. It is suggested that disrupted neuroplasticity can cause treatment resistance and difficult-to-treat symptoms such as ruminations and anhedonia [28]. One of the markers of response in the treatment of depression is the hippocampal volume. A study showed that a lower baseline hippocampal volume is related to bad treatment response [34]. A systematic meta-analysis of response predictors found that a smaller right hippocampal volume is related to less improvement in depression [35]. Another study found increased grey matter volume in the amygdala in resistant versus non-resistant patients with depression [36]. According to a voxel-based morphometric study, patients who did not achieve remission had significantly reduced grey matter volume in the left dorsolateral prefrontal cortex (DLPC) compared to remitters and to the control group [37]. 

In addition to structural changes in the course of resistant depression, functional disturbances such as abnormal communication between anatomically separate regions are typically investigated. This is observed in studies using resting-state functional imaging (Rs-fMRI). A recent systematic review of neuroimaging studies in TRD found alterations in the default mode network (DMN) connectivity [38]. The DMN constitutes a functional network that is active during spontaneous cognition and involves processing internally directed information such as the persons’ past and future and analyzing their mental states [39]. According to the data from the systematic review, hypoconnectivity of the DMN with other brain regions, as well as decreased functional connectivity of various brain regions with DMN, is specific for treatment resistance. The authors state that altered functional connectivity between DMN and sensory areas of the brain in TRD patients may be the reason for the disrupted integration of sensory content concerning positive life events. Besides hypoconnectivity of the DMN, they found that hyperactivity in cortical regions of the DMN is also specific for TRD and can be related to increased rumination. The authors conclude that DMN alterations can differentiate resistant patients from responsive ones. They also suggest that TRD is related more to functional than structural abnormalities [38]. Earlier meta-analysis revealed that inter-network functional connectivity (FC) disturbance is related to psychomotor retardation [39]. A recent study found that connectivity disruptions differ in patients with first depression episodes and with recurrent depression. FC changes were more pronounced in patients with subsequent episodes and some changes were still observed in the remission phase. It also seems that the quality of functional connectivity impairment depends on the illness duration. These preliminary results suggest that connectivity changes have a progressive nature [40]. 

A large multicenter study with the functional magnetic resonance imaging (fMRI) technique allowed for the distinguishing of four biotypes of depression characterized by distinct patterns of disturbed connectivity in the limbic and frontostriatal circuits. These biotypes predict a response to transcranial magnetic stimulation (TMS) [41]. A study on functional connectivity in the frontopolar cortex (FPC) suggests that disturbed connectivity in this subregion may cause dysfunction of the reward system and cognitive functions, thus causing a lack of response to treatment [42]. An interesting branch of PET (positron emission tomography) studies investigates microglial activation in depression. It is suggested that microglial overactivity is related to poor response, longer disease duration, and increased inflammatory response [43]. A recent preliminary study combined PET and magnetic resonance imaging (MRI) and found a connection between disturbed connectivity and increased microglial activity in MDD [44].

Another group of biomarkers of treatment-resistant depression is inflammatory cytokines. Inflammation is linked to depression and is likely one of the factors worsening treatment response. Peripheral cytokines activate inflammatory mediators in the brain. They affect neurotransmitter transport and BDNF production and disturb neuroplastic processes. Proinflammatory cytokines such as interleukin-1 (IL-1), interferon gamma (INF-γ), and tumour necrosis factor-alpha (TNF-α) affect the glutamatergic system through activation of the kynurenine pathway, which contributes to excitotoxicity through reducing tryptophane bioavailability and the production of quinolinic acid, which subsequently overstimulates N-methyl-D-aspartate (NMDA) receptors [45]. Monoamine and neuropeptide systems are also affected [46]. A recent meta-analysis has shown that elevated C-reactive protein (CRP), interleukin-6 (IL6), and (TNF)-alfa contribute to treatment resistance [47]. Peripheral C-reactive protein (CRP) concentration seems to be higher in TRD than in MDD patients [48]. A recent systematic review found that IL-6 and CRP/high sensitivity CRP could help predict treatment response in TRD [49]. Another meta-analysis found that lower baseline interleukin-8 (IL-8) levels were associated with treatment response. Interestingly, IL-8 is produced in the central nervous system mainly by microglia and through a positive feedback mechanism; this cytokine sustains the inflammatory response [50].

### 2.3. Ketamine in Treatment-Resistant Depression

#### 2.3.1. Ketamine Enhances Synaptic Plasticity

Traditionally acting antidepressants (TAA) have limited effectiveness and there is a significant group of patients who do not reach remission or even response. According to the above-mentioned STAR*D trial, one-third of patients achieved remission following initial antidepressant medication [51]. The STAR*D trial has also shown that fewer than 15% of patients who failed two anti-depressants were able to reach remission in each of the next two treatments steps, and 90% of patients who do not respond to two treatment trials become treatment resistant. [Rush 2006] A Treatment-Resistant Depression Cohort in Europe study found that 55% of patients experienced two treatment failures and 30% had three unsuccessful drug trials. After 6 months, 73% of patients did not show a response. After 1 year, 19.2% were remitted and 69% showed no response [9].

There is a definite need for new strategies involving different mechanisms of action in TRD [52]. Ketamine, an NMDA receptor antagonist, caused a breakthrough in antidepressant treatment. It has a rapid antidepressant and anti-suicidal effect in patients with unipolar and bipolar depression after a single infusion, although more stable improvement is observed after multiple administrations [53,54,55]. Based on its efficacy and safety, [56] ketamine’s s-enantiomer has been approved by the FDA in an intranasal form as an add-on medication to an antidepressant in TRD [57]. 

There are three hypotheses explaining ketamine’s rapid antidepressant action. The first one involves the disinhibition of glutamate through an NMDA blockade in inhibitory neurons. A glutamate surge activates α-amino-3-hydroxy-5-methyl-4-isoxazole propionic acid receptors (AMPA) and causes the release of BDNF and mTOR (mechanistic target of rapamycin), which increases the number and function of synapses in vitro [58]. The activation of the AMPA receptor stimulates serotonin neurotransmission through descending cortico-raphe projections, adding to the antidepressant effect [59]. The second mechanism involves intracellular signaling. The NMDA blockade stops calcium flow into the neuron, which causes inhibition of eukaryotic elongation factor 2 kinase (eEF2K), the dephosphorylation of eukaryotic elongation factor 2 (eEF2), and an increase in BDNF translation which allows rapid production of this neurotrophin in the hippocampus [33]. The third hypothesis involves the direct binding of ketamine to the TrkB receptor [30]. Studies investigating molecular neuroplastic mechanisms behind the rapid antidepressant and neuroplastic effect of ketamine are included in a recent systematic review and provide a basis for the above hypotheses [60]. The described mechanisms are presented in Figure 2.

Traditionally acting antidepressants also increase BDNF, but they do it slowly and do not cause the activity-dependent release that is necessary for synaptic plasticity. In other words, they increase BDNF expression, but not release [58,61]. The effect of TAA involves long-term processes based on translational modifications and chromatin remodeling [62]. Ketamine starts acting within hours, not weeks like TAAs. It rapidly increases synaptogenesis in the medial prefrontal cortex (mPFC) and reverses deficits caused by exposure to chronic stress in rodents [63,64]. This observation suggests that ketamine could improve the synaptic deficits and disturbed connectivity caused by depression [31]. One study showed that ketamine induces spine formation in the prefrontal cortex and improves microcircuits’ activity in vivo in mice. Newly formed spines seem to be necessary to sustain the antidepressant effect over time. The authors suggest that following pharmacological and neurostimulatory interventions, preserving the rescued synapses may be needed to prolong remission [65]. An interesting result was found in an animal study with a TRD model where animals that did not respond to ketamine received lithium augmentation and responded with enhanced coping behavior under stress and where lithium alone was ineffective [66]. Increased insulin signaling in the augmentation group was observed. The authors suggest that lithium augmentation of ketamine may be beneficial only in the case of a deficit in certain response moderators, such as insulin signaling, and may enable antidepressant responsivity in patients who do not respond to ketamine [66]. This kind of perspective can change the understanding of TRD and confirms the need for an individualized approach in future studies, giving hope for developing precise psychiatry strategies.

According to a recent review, ketamine evokes a unique form of functional neuroplasticity called homeostatic plasticity, which resembles homeostatic synaptic scaling. It uses negative feedback as a response to deviations from specific activity patterns and brings neuronal activity back to its initial state [67]. Homeostatic plasticity may be one of the ways ketamine compensates for circuit dysfunction and activates processes that are not normally present [30]. Ketamine’s metabolite (2R,6R)-hydroxynorketamine (HNK) is hypothetically responsible for the long-lasting effect of ketamine. It increases BDNF levels in the hippocampus and has an antidepressant effect, although its binding site has not been discovered thus far [31]. 

Studies show that apart from BDNF, VEGF also plays an important role in the antidepressant action of ketamine and adds to its effect on neuroplasticity. Ketamine rapidly increases VEGF release in mPFC pyramidal neurons in mice and causes an increase in the number and function of synapses [31,68].

#### 2.3.2. Ketamine Affects Brain Volume

More evidence for the neuroplastic properties of ketamine comes from imaging studies. A randomized controlled structural MRI study on healthy volunteers showed that subfield hippocampal volumes were significantly larger post single ketamine infusion vs placebo, confirming, indirectly, ketamine’s short-term neuroplastic effect in humans [69]. Another recent study on MDD patients who received six ketamine infusions investigated the change in the volume of specific subcortical structures and hippocampal subfields. The volume increase was observed in the right hippocampus and left amygdala subfields. The authors also found that increases in the volume of the right thalamus and left subiculum head in the hippocampus can predict a better response in MDD [70]. An earlier study tested MDD patients using high-resolution MRI to estimate the volume of nucleus accumbens (NAc) following single ketamine infusion. Longitudinal data from this study, although preliminary, suggest normalization, meaning a reduction in left NAc and an increase in left hippocampal volumes in patients who achieved remission following ketamine administration [71]. Some studies did not find any correlation between baseline subcortical volumes and a single infusion of ketamine [72].

#### 2.3.3. Ketamine Alters Functional Connectivity

Another piece of evidence for ketamine’s neuroplastic properties comes from the observation of its effect on functional connectivity in the human brain. One study using resting-state functional connectivity MRI (rs-FC MRI) investigated global brain connectivity with global signal regression (GBCr) changes 24 h after a single ketamine infusion in unmedicated MDD patients vs control. The study revealed reduced GBCr in patients with MDD compared to healthy subjects, and a significant increase in GBCr in the PFC was observed 24 h after ketamine infusion. Response to ketamine correlated with the GBCr increase in the lateral PFC, caudate, and insula. The authors state that specific connectivity changes could be a biomarker for treatment response. According to the authors, a few questions remain unanswered, such as whether GBCr reduction is specific for MDD and for ketamine, what the connection between the GBCr pattern and synaptic dysconnectivity is, and what the long-term effects of ketamine on GBCr are [71]. A more recent study investigated resting state FC within an affective network and DMN in patients with TRD in the course of MDD or bipolar disorder before and after four ketamine infusions administered over two weeks. Additionally, ketamine patients received lithium or a placebo. The study demonstrated that resting state FC changes between the amygdala and subcallosal/subgenual anterior cingulate cortex in the right hemisphere at baseline predicted the response to ketamine treatment irrespective of the lithium or placebo [73]. Another recent study explored resting-state FC changes in habenula and DMN in 65 patients with depressive episodes due to MDD or bipolar disorder before and after six open-label ketamine infusions as an add-on treatment. The authors found decreased connectivity between the right habenula and bilateral angular cortex in responders. There were no alterations observed in non-responders. The study results suggest that ketamine may suppress connectivity between habenula and DMN producing an antidepressant effect [74]. Considering the above-mentioned results, which suggest the progressive nature of FC changes in recurrent depression, it can be hypothesized that ketamine, by affecting the FC, can have a transformative role in TRD [40].

#### 2.3.4. Ketamine Has an Immunomodulatory Effect

The activation of the immune system is associated with depression and the anti-inflammatory effect of antidepressant treatment is related to response [75]. A recent meta-analysis revealed that antidepressant treatment was related to a significant TNF-α decrease in responders [50]. A previous meta-analysis also found a correlation between a higher TNF-α level and treatment resistance. Another result of this meta-analysis was an IL-6 decrease related to treatment, but not the response [47]. As mentioned before, activation of the kynurenine pathway by inflammatory cytokines causes an increase in the production of quinolinic acid, activates NMDA receptors, and induces cytotoxicity. Therefore, targeting this process could inhibit this effect. Ketamine not only reduces peripheral cytokine levels, but also interrupts the kynurenine pathway regulating the hyperglutamate state and increasing serotonin levels, which are linked to its anti-suicidal properties [76]. It also alters and decreases macrophage expression of TNF-α, IL-6, and IL-1β, and influences microglia and astrocytes [77]. 

A study using a rodent model of depression investigated microglial activity after a single ketamine infusion, then, based on the results, it explored the kynurenic acid to quinolinic acid ratio as a potential marker of the response in patients with TRD after multiple ketamine infusions. The murine part of the study revealed the antidepressant effect of ketamine, a decrease in parenchymal cytokine production, and the quinolinic acid production by microglia, which correlated with the peripheral level. The clinical part showed that quinolinic acid could be used as a ketamine response marker in patients with depression. The authors also state that microglia are a key therapeutic target for antidepressant treatment [77]. 

Another target for ketamine is astrocytes, cells that produce BDNF and have many NMDA receptors. Astrocyte activation may play an important role in the antidepressant effect of ketamine. An animal study investigated fibrillary acidic protein expression as an indicator of astrocyte activity after single ketamine vs scopolamine administration. Ketamine, contrary to scopolamine, increased astrocyte activity in the hippocampus and astrocyte inhibition abolished the antidepressant effect of ketamine but not scopolamine [78]. Generally, in vitro and animal studies confirm the anti-inflammatory effect of ketamine as a part of its antidepressant action, but there is still a need for more clinical studies in this field. 

Neuroplastic mechanisms of ketamine are presented in Figure 3.

#### 2.3.5. Ketamine Enhances Resilience

According to the latest research, ketamine, probably through its neuroplastic effects, enhances stress resilience, and may prevent depression due to exposure to stress. Stress resilience has been defined as the ability to experience stress without developing psychopathology. Stressful life events constitute a risk factor for developing treatment-resistant depression [79]. A rodent study found that ketamine administration 1 h before stress correlated with fewer hedonic deficits and significant differences in dendritic spine density and morphology in CA3 subregions of the hippocampus compared to the control [80]. Preliminary evidence in women administered with a single dose of ketamine epidurally shortly after a caesarean cut suggests that ketamine may reduce the risk of developing depressive symptoms after acute stress and postpartum depression. It has been proposed that ketamine could be used as a prophylaxis for soldiers before participation in combat [81]. The mechanism behind this effect is still unknown, but it likely involves the glutamatergic system, mesolimbic dopamine pathway, PFC, and the hippocampus, and possibly the anti-inflammatory effect of ketamine [81]. 

## 3. Discussion

Staging in TRD, as was already mentioned, comes from oncology. As in the case of cancer, in TRD, the stage of the disease accounts for the prognosis, and the chance for improvement decreases with the stage of progression. Following this analogy, we set a concept of downstaging in TRD. The downstaging phenomenon is established in oncology research. An example is the remarkable effect of neoadjuvant therapy, which describes a situation in which the effect of treatment allows for the definition of the stage of the disease as less advanced [82,83,84]. Similarly, in heart failure, where New York Heart Association (NYHA) functional class is a commonly used staging method, novel treatment such as sacubitril/valsartan can cause a reduction in the NYHA class [85,86]. We propose a provocative hypothesis that in treatment-resistant depression ketamine, in combination with other treatments, could be considered as such a neoadjuvant therapy and allow for formal disease stage reduction. We hypothesize that a group of TRD patients can respond to ketamine treatment by downstaging. The transformative effect of ketamine could hypothetically increase the functional reserve and reduce the level of treatment resistance. It could help to achieve meaningful improvement, not only in clinical symptoms, but also in psychosocial functioning from patients’ own perspectives [19]. A very recent systematic review investigating ketamine’s antidepressant effect depending on the stage of TRD revealed that although ketamine is efficacious in all patient groups, it has a smaller effect in more advanced TRD stages [87]. This is in line with the suggestion from the review by McAllister et al. that therapies should be considered earlier in treatment-resistant patients [88]. The great challenge for future research is identifying this patient subgroup. Although provocative, it may be a useful paradigm to implement in future research with patient benefit prioritized above all. Such a concept can change the way patients and psychiatrists look at chances for improvement. It could modify the focus on treatment outcomes and hopefully encourage psychiatrists to implement more advanced treatment strategies sooner in order to dampen the severity of the disease. This approach needs confirmation in longitudinal studies assessing not only depressive symptomatology, but also the level of functioning in different areas of life. They should also consider patients’ perspectives; therefore, patient report outcomes should be measured. Particularly interesting would be how patients perceive the treatment and its short- and long-term effects, as well as what their hopes and fears are [89]. The downstaging concept is presented in Figure 4.

## 4. Materials and Methods

We have included papers available up to September 2022 using PubMed and Web of Science. The search terms were ‘treatment-resistant depression’, ‘depression’, ‘difficult-to-treat depression’, ‘staging’, ’downstaging’, ‘ketamine’, ‘esketamine’, ‘arketamine’, ‘neuroplasticity’, ‘synaptic plasticity’, ‘BDNF’, ‘hippocampal volume’, ‘microglia’, ‘inflammation’, ‘functional connectivity’, ‘default mode network’ ‘immunomodulatory’, ‘kynurenine pathway’, ‘antidepressants’. We selected mainly human studies based on their high methodological quality, and on how informative and innovative they were. We included not only original research but also reviews.

## 5. Conclusions

Treatment-resistant depression is a severe and debilitating disease that can be defined by stages in the disease’s progression. Here, based on recent data on ketamine’s mechanisms of action in treatment-resistant depression, we present the concept of downstaging. We hope this new perspective will encourage clinicians and researchers to engage in treatment and studies, hopefully relieving patients from suffering.

## Figures and Tables

**Figure 1 ijms-23-14605-f001:**
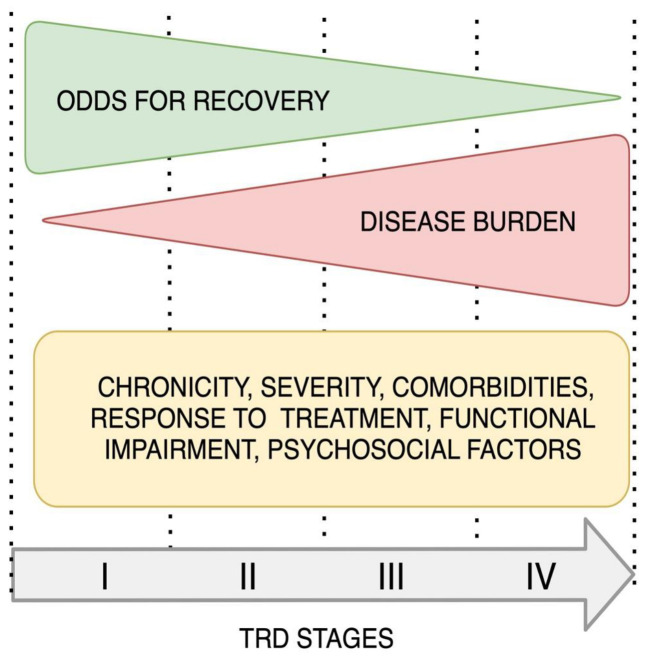
The progressive course of TRD. Modified from Fekadu et al. 2009, van Belkum et al. 2018. TRD (treatment-resistant depression) [24,25].

**Figure 2 ijms-23-14605-f002:**
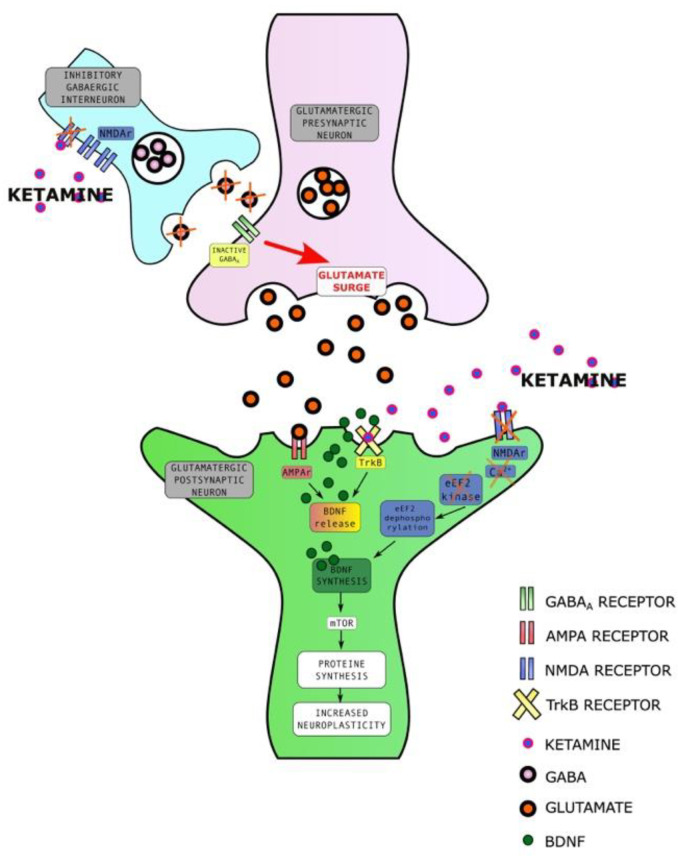
Molecular mechanisms of rapid antidepressant and neuroplastic effect of ketamine. NMDAR—N-methyl D-aspartate receptor, GABA_A_ R—gamma-aminobutyric acid A receptor, AMPAR—α-amino-3-hydroxy-5-methyl-4-isoxazolepropionic acid receptor, BDNF—brain derived neurotrophic factor, m-TOR—mammalian target of rapamycin, Ca^2+^—calcium ions, EF2K—eukaryotic elongation factor 2 kinase, EF2—eukaryotic elongation factor 2, TrkB—tyrosine receptor kinase B.

**Figure 3 ijms-23-14605-f003:**
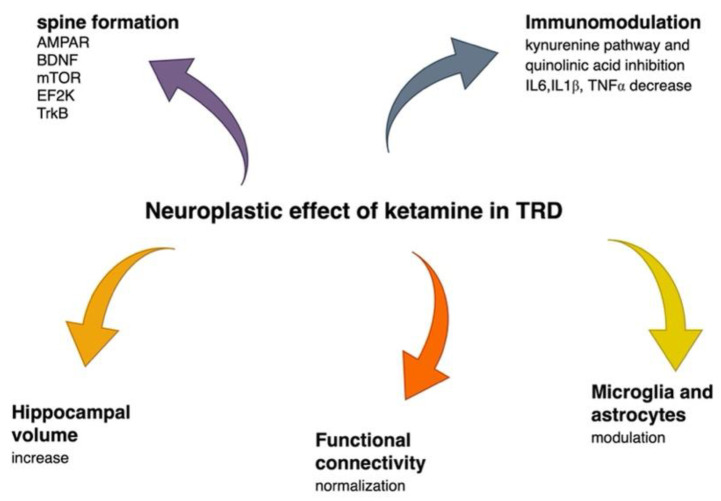
Ketamine’s neuroplastic effect in treatment-resistant depression. AMPAR—α-amino-3-hydroxy-5-methyl-4-isoxazolepropionic acid receptor, BDNF—brain derived neurotrophic factor, m-TOR—mammalian target of rapamycin, EF2K—eukaryotic elongation factor 2 kinase, TrkB—tyrosine receptor kinase B, IL6—interleukin 6, IL1ß interleukin 1 beta, TNF α—tumor necrosis factor α.

**Figure 4 ijms-23-14605-f004:**
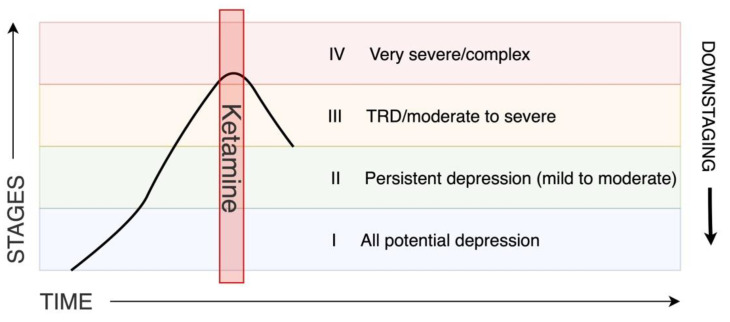
Downstaging concept. Stepped care pathway for depression according to NICE guideline 2009 [26]. TRD (treatment-resistant depression).

## Data Availability

Not applicable.

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
