# Peer review of "The Downstaging Concept in Treatment-Resistant Depression: Spotlight on Ketamine"

_ijms, 2022, doi:10.3390/ijms232314605_

Round 1
Reviewer 1 Report
Interesting paper about a narrative review on the concept of downstaging in treatment-resistant depression, highlighting the use of ketamine in this situation.
Corrections to be made: On line 22 it should be written "... mood disorders (MDD)..." and on line 23 "... treatment resistant depression (TRD)..." On line 174 what is the "PFC "? On line 217 what is the "FC"? On line 224 what is "fMRI"? On line 233 what is "MRI"? And many other acronyms are not explained when first used in the text.
Author Response
Dear reviewer thank you very much for your suggestions, we have explained all the acronyms when first used.
Reviewer 2 Report
The manuscript is well written. But I suggest including a table showing the studies used for ketamine and BDNF and their signalling pathway.

Author Response
Dear reviewer, thank you for your valuable suggestion, we started doing the table, but then we found a very recent, excellent systematic review (Kang MJY, Hawken E, Vazquez GH. The Mechanisms Behind Rapid Antidepressant Effects of Ketamine: A Systematic Review With a Focus on Molecular Neuroplasticity. Front Psychiatry. 2022,25;13:860882. doi: 10.3389/fpsyt.2022.860882.) including all those studies, so we decided to cite it and instead of a table we drew another figure (Figure 2) illustrating all the ketamine neuroplastic pathways which are described in the text for better clarity.
Reviewer 3 Report
In this manuscript, the authors propose a model of staging in the conceptualization of treatment resistant depression, and propose the use of ketamine as an additional treatment for those with TRD. Overall, the authors do a fine job in describing the staging model and reviewing the current literature in the potential for ketamine as a therapeutic agent for treatment resistant depression. I have only a few minor comments.
1) The manuscript is well written overall and the authors do a fine job in thoroughly explaining the staging model and literature regarding ketamine. It would be helpful though to have a native English speaker proofread the manuscript to aid in clarifying some minor wording and phrasing issues.
2) Where appropriate, the authors should use person-first terminology when describing people with major depression. For example, the end of the first sentence of the abstract could be revised to "30% of patients with depression" from "30% of depressed patients."
3) Abbreviations should be defined when first used and used consistently throughout the manuscript.
Author Response
Dear reviewer, thank you for your kind remarks, we have asked for professional editing and a native speaker checked the manuscript. We have corrected the sentence from the abstract according to your suggestion, we also checked the whole manuscript and used person-first terminology when patients were mentioned. We have also carefully checked all the abbreviations, thank you.